# Rational Design of Oxazolidine-Based Red Fluorescent pH Probe for Simultaneous Imaging Two Subcellular Organelles

**DOI:** 10.3390/bios12090696

**Published:** 2022-08-29

**Authors:** Chunfei Wang, Hengyi Fu, Jingyun Tan, Xuanjun Zhang

**Affiliations:** 1Faculty of Health Sciences, University of Macau, Macau 999078, China; 2MOE Frontiers Science Centre for Precision Oncology, University of Macau, Macau 999078, China

**Keywords:** pH-responsive probe, reversible detection, mitochondria and lysosomes, simultaneous colocalization

## Abstract

A reversible pH-responsive fluorescent probe, BP, was rationally designed and synthesized, based on protonation and deprotonation gave rise to oxazolidine ring open and close. The fluorescence response of BP against pH ranges from 3.78 to 7.54, which is suitable for labeling intracellular pH-dependent organelles. BP displayed strong red emission at a relatively high pH in living HeLa cells and U87 cells. More importantly, this probe exhibited good colocalization with both mitochondria and lysosomes in these two cell lines, attributing to pH-induced structure tautomerism resulting in an oxazolidine ring open and close that triggered effective targeting of these two organelles. As organelle interactions are critical for cellular processes, this strategy of targeting dual organelles through the structure tautomerism is conducive to further developing more effective and advanced probes for real-time imaging of the interaction between mitochondria and lysosomes.

## 1. Introduction

Acids and bases exist a dynamic balance in living organisms, whereas pH value effectively reflects the level between acids and bases in cells, body fluids and organs [1]. Intracellular pH ranges from 4.5 in lysosomes to 8.0 in mitochondria [2,3]. Due to the requirement for unique pH values in different regions and organelles, intracellular pH can provide cells with a special circumstance closely related to their physiological activity [3,4,5,6]. Moreover, pH plays a decisive role in many biochemical procedures, including metabolism, ion transport, activation of enzymes, and the like [7,8,9,10,11]. Nevertheless, pH abnormalities will disrupt proper cell functions, which is considered a hallmark in some pathophysiological processes, such as inflammation, shock and cancer [12,13]. Consequently, there is tremendous interest in monitoring abnormal pH fluctuations in living cells to understand pathological effects and physiological activities, as well as for the early diagnosis of diseases. 

Dynamic organelle interactions regulate intracellular signaling outcomes and it thereby has a close relationship with cell survival [14,15]. Mitophagy, known as the removal of excess and damaged mitochondria [16,17,18], is a typical paradigm of mitochondria and lysosome interactions. Additionally, mitochondria–lysosome interactions, also involving fusion between mitochondria and lysosomes and mitochondria–lysosome contact (MLC) [19,20,21,22], are important intracellular processes in eukaryotic cells. However, dysregulated MLC is also linked to diseases such as Parkinson’s disease [23,24,25]. Fluorescent probes with precise imaging capabilities have been widely used in biological-chemical analyses and have some superiorities in monitoring intracellular and extracellular pH [5,26,27]. Recently, small molecular fluorescent probes have been successfully employed for sensing intracellular pH or pH in different organelles (such as cell nuclei, lysosomes and mitochondria) [28,29,30]. For acidic lysosomes and alkalescent mitochondria, real-time monitoring by only one fluorescent pH probe can obtain deep insight into the more complicated and important cellular or subcellular activities, as well as their interactions. Many recent studies have made great success in exploring the contact between lysosomes and mitochondria, as well as a useful biomolecule HClO containing [31,32,33,34]. To understand the interactions more fully between mitochondria and lysosomes, fluorescent pH probes can be utilized to simultaneously image them, considering their difference in pH, but there is a lack of powerful tools at present. 

Pyridyl nitrogen atoms and phenolic oxygen atoms are frequently utilized as protonated targets in the fluorescent sensing of pH [35,36]. Furthermore, large numbers of supramolecular functional systems have also been implemented to track pH [37]. Currently, a novel fluorescent probe, BP, has been synthesized in terms of pH-dependent equilibria between the ring-opening and ring-closing isomers of oxazolidine. Significantly, the hydroxypropyl group tends to form ring-closed oxazolidine in BP at a relatively high pH, with strong red fluorescence (Figure 1A). Moreover, BP was also successfully applied for real-time monitoring of pH in living HeLa cells and U87 cells. In particular, BP displayed good colocalization with both mitochondria and lysosomes in these two cell lines, which was attributed to the formation of specific organelle-targeting structures (Figure 1B). This study provides a new strategy for simultaneously tracking different organelles in real-time cellular visualization. It is anticipated to be rationally applied in further designing more effective probes for the study of organelle interactions, especially mitochondria–lysosome interactions. 

## 2. Materials and Methods 

### 2.1. Materials, Instruments and General Methods

Chemicals containing 2,4-dimethyl-3-ethylpyrrole, 2,3,3-trimethyl-3*H*-indole, 2-bromoethanol and other analytes (NaCl, KCl, CaCl_2_ and MgSO_4_, ect.) were provided by Dieckman (Hong Kong, China) Chemical Industry Company Ltd. (Hong Kong, China). Fetal bovine serum (FBS) was purchased from Gibco/BRL (Grand Island, NE, USA). Dulbecco’s modified eagle medium (DMEM) and other chemicals (dimethylsulfoxide, DMSO) were obtained from Sigma-Aldrich (St. Louis, MO, USA). MitoTracker Green FM, LysoTracke Green DND-26 and LysoTracker Blue DND-22 were purchased from ThermoFisher (Waltham, MA, USA). All other reagents were AR grade and used without further purification. Distilled water was purified using a Milli-Q water purification system provided by Millipore (Bedford, MA, USA).

NMR spectra detection was recorded by a Bruker AV-400 instrument (Bruker, Karlsruhe, Germany) and reported in ppm downfield. High-resolution mass spectra (HRMS) were obtained by an Xevo G2-XS QTof spectrometer (Waters, Milford, MA, USA). All tested compounds were determined through a reverse phase C_18_ column by peak area integration (Agilent HPLC 1260, Palo Alto, CA, USA). UV/Vis absorption and fluorescent spectroscopy were measured with a UV spectrophotometer (UV-1800, Shimadzu, Kyoto, Japan) and fluorescence spectrophotometer (FluoroMax-4, Horiba, Kyoto, Japan). Cell imaging was evaluated with confocal fluorescence microscopy (Carl Zeiss 710, Jena, Germany).

### 2.2. Synthesis of pH Probe BP

As described in Figure 2, BOBPY-CHO was synthesized according to a previous study [38,39]. The synthesis of intermediates **1** and **2** was performed with a previous study for reference [40]. A mixture of 2,3,3-trimethyl-3*H*-indole (20 mmol, 3.06 g) and 2-bromoethanol (25 mmol, 3.12 g) in acetonitrile (20 mL) was refluxed for 24 h under N_2_. After cooling to room temperature, the solvent was removed using rotary evaporators. The residue was suspended in hexane (25 mL), sonicated and filtered. The crude solid was recrystallized from chloroform (35 mL) to afford **1** (yield 65.35%). ^1^H NMR (400 MHz, DMSO) δ = 7.97 (dd, J = 5.6, 3.4, 1H), 7.88−7.83 (m, 1H), 7.65−7.59 (m, 2H), 4.67−4.55 (m, 2H), 3.94−3.82 (m, 2H), 2.83 (s, 3H), 1.55 (s, 6H).

A solution of **1** (10 mmol, 2.83 g) and KOH (15 mmol, 0.84 g) in H_2_O (40 mL) was stirred at room temperature for 10 min and then it was extracted with diethyl ether (3 × 20 mL). The organic phase was concentrated under reduced pressure to afford **2** (yield 90.00%) as yellow oil. ^1^H NMR (400 MHz, Acetone) δ = 7.11−7.05 (m, 2H), 6.84 (td, J = 7.5, 1.0, 1H), 6.77 (dd, J = 8.2, 0.8, 1H), 3.77 (dddd, J = 15.1, 12.0, 7.3, 3.0, 2H), 3.54 (dt, J = 12.0, 8.5, 1H), 3.42−3.35 (m, 1H), 1.37 (s, 3H), 1.32 (s, 3H), 1.13 (d, J = 4.1, 3H). ^13^C NMR (101 MHz, Acetone) δ = 151.20, 139.91, 127.23, 122.11, 121.08, 111.68, 108.49, 62.55, 49.48, 46.57, 27.35, 20.16, 17.76. 

The synthesis of the pH probe, BP, was in accordance with a previous report [41]. Trifluoroacetic acid (TFA, 30 mmol) was added dropwise to a solution of **2** (5 mmol, 1.02 g) and BOBPY-CHO (6 mmol, 2.74 g) in ethanol (10 mL). The mixture was refluxed for 24 h. After cooling to room temperature, the solvent was removed under reduced pressure and the residue was dissolved in dichloromethane (5 mL). The addition of diethyl ether (200 mL) and refrigeration for 12 h caused the formation of a precipitate. After filtration, the solid residue was dissolved in aqueous NaHCO_3_ (5% w/v, 20 mL) and stirred for 1 h at ambient temperature. The aqueous mixture was extracted with ethyl acetate (3 × 40 mL). The organic phase was dried over anhydrous Na_2_SO_4_, filtered and the solvent was distilled off under reduced pressure to give BP (yield 40%). ^1^H NMR (400 MHz, CDCl_3_) δ = 8.02 (d, J = 8.3, 1H), 7.90−7.84 (m, 1H), 7.80 (dd, J = 8.1, 0.6, 1H), 7.41−7.34 (m, 1H), 7.31 (d, J = 1.2, 1H), 7.27−7.01 (m, 8H), 6.95 (d, J = 7.0, 1H), 6.88 (t, J = 7.5, 1H), 6.82 (t, J = 7.4, 1H), 6.64 (dd, J = 14.1, 11.9, 2H), 6.03 (dd, J = 15.9, 1.7, 1H), 3.59−3.46 (m, 3H), 3.31−3.19 (m, 1H), 2.40 (d, J = 11.7, 3H), 2.36−2.23 (m, 2H), 2.19 (s, 3H), 1.28 (s, 3H), 1.19 (d, J = 7.0, 3H), 0.98−0.96 (m, 3H). ^13^C NMR (101 MHz, CDCl_3_) δ = 157.00, 150.92, 150.66, 150.62, 143.26, 139.79, 139.76, 135.34, 134.38, 133.44, 133.10, 132.84, 132.79, 132.05, 131.93, 130.73, 130.41, 128.77, 127.49, 126.38, 125.46, 124.12, 123.43, 122.35, 120.19, 119.78, 116.01, 109.99, 109.96, 63.40, 63.37, 50.00, 49.94, 47.82, 47.78, 29.72, 29.34, 28.27, 17.49, 14.86, 14.22 13.13, 9.61. HRMS (ESI): calcd. for C_43_H_40_BN_3_O_2_ [M+H]^+^, 642.3292; found 642.3293.

### 2.3. Preparation of the Test Solution 

A stock solution (10 mM) of BP was prepared in DMSO. Then, test solutions of BP were diluted to 2 mL with a mixture of acetonitrile and Britton-Robinson (B-R) buffer solution (*v*/*v* = 1:1) with different pH values, which were used to detect UV-vis absorption and fluorescence properties.

The solutions of various analytes were prepared from NaCl, KCl, CaCl_2_, MgCl_2_, FeCl_3_, CuCl_2_, ZnCl_2_, NiCl_2_, MnCl_2_, CoCl_2_, AlCl_3_, arginine, glutamic acid, cysteine, glutathione, H_2_O_2_ and NaClO, respectively. Small aliquots of each testing species solution were added. The resulting solution was shaken well and incubated for 30 min at room temperature before the fluorescent spectra were recorded.

### 2.4. Cell Viability Detected by MTT 

Human cervical cancer HeLa cells and human glioma U87 cells were cultured in a complete DMEM culture medium containing 10% FBS and 1% penicillin-streptomycin at 37 °C in an atmosphere containing 5% CO_2_. After 90% confluence, the cells were cultured in 96-well plates (5000 cells/well, 100 μL). The cells were treated with different concentrations of BP (5, 10, 20, 30, 40, 50, 80 μM) for 24 h. Then, an MTT stock solution (5.0 mg/mL, 100 μL) was added to each well and incubated for 4 h to form a purple crystal formazan. At the end of incubation, DMSO (100 μL) was added, followed by 10 min of microvibration after removing the medium. Absorbance at 570 nm was measured using a microplate reader (Thermo, Waltham, MA, USA).

### 2.5. Cell Treatment and Imaging

After 90% confluence, HeLa cells and U87 cells were seeded in culture dishes at a density of 1 × 10^5^ for 24 h. Before removing the medium, HeLa cells or U87 cells were treated with 10 μM BP for 2 h and washed with PBS buffer (pH = 7.4) three times. Then, HeLa cells and U87 cells were randomly divided into 3 groups and incubated with B-R buffer at different pH values (pH = 3.78, 5.33 and 7.54) for 10 min. After the culture medium was discarded, the cells were fixed. Images were collected by confocal microscopy (Carl Zeiss LSM710, Jena, Germany).

For colocalization of lysosomes and mitochondria, the cells were pretreated with 20 μM MitoTracker Green FM (λ_ex_ = 490 nm and λ_em_ = 513 nm) or 20 μM LysoTracker Green DND-26 (λ_ex_ = 504 nm and λ_em_ = 511 nm) for 30 min and then washed with PBS buffer (pH = 7.4) three times. After that, the cells were treated with 10 μM BP for 1 h. Before imaging, the culture medium was discarded, and the cells were washed with PBS buffer (pH = 7.4) three times and then fixed. Images were also collected by confocal fluorescence microscopy (Carl Zeiss LSM710, Jena, Germany). Data from three independent experiments were then analyzed using Image J. 

In addition, the two cell types were pretreated with 20 μM MitoTracker Green FM (λ_ex_ = 490 nm and λ_em_ = 513 nm) and 20 μM LysoTracker Blue DND-22 (λ_ex_ = 373 nm and λ_em_ = 422 nm) for 30 min. After washing with PBS buffer (pH = 7.4) three times, the cells were subsequently incubated with 10 μM BP for 1h. Before imaging, the culture medium was discarded, and the cells were washed with PBS buffer (pH = 7.4) three times and then fixed. Images were also collected by confocal fluorescence microscopy (Carl Zeiss LSM710, Jena, Germany). Data from three independent experiments were then analyzed using Image J.

## 3. Results and Discussion

### 3.1. Design and Synthesis of BP

N_2_O-type benzopyrromethene boron complexes (BOBPYs) were first developed by Jiao and coworkers [39], wherein different boronic acid derivatives were substituted in the axial position, resulting in high stability, strong absorbance, high fluorescence quantum yields, and narrow emission bands. In addition, N-Hydroxypropyl indole derivatives display oxazolidine ring opening and closing induced by light, temperature and other chemical stimulations (acid-base) [6,40,41]. Nowadays, it has been extensively adopted for pH sensors. BOBPY-CHO, whose axial position was substituted with 4-formylphenylboronic acid, was flexible for further functionalization. Subsequently, the pH probe BP was synthesized by a Knoevenagel reaction with BOBPY-CHO. The detailed synthetic procedures for BP are described in experimental Section 2.2. All products were fully characterized by ^1^H NMR, ^13^C NMR and HRMS spectra (Appendix A). Under acidic conditions, a BOBPY-oxazolidine π-conjugated system was displayed, whereas deprotonation triggered oxazolidine ring closing, and π-conjugation was disturbed, resulting in strong red fluorescence in the BP probe.

### 3.2. Absorption and Fluorescence Response of BP

The absorption and fluorescence spectra of BP were measured in a mixture of MeCN and Britton-Robinson (B-R) buffer solution (*v*/*v* = 1:1) with different pH values. As shown in Figure 1A, BP possessed the main absorption peaks at 400 nm and 620 nm. The characteristic absorption peaks at 400 nm decreased with increasing pH from 3.78 to 7.54, which was induced by the opening and closing of the oxazolidine ring. However, the absorption peaks at 620 nm displayed almost no change. BP exhibited a very weak fluorescence peak at 630 nm in the mixture of MeCN and BR buffer solution (*v*/*v* = 1:1) when the pH was 3.78, while the peaks increased at pH 7.54, with a 25-fold intensity enhancement (Figure 1B). When the pH was 3.78, the oxazolidine ring opened and iminium ions (C=N^+^) were immediately formed, resulting in fluorescence quenching because of the existence of a BOBPY-oxazolidine π-conjugated system. As the pH increased, BP was more likely to have a closed ring, exhibiting strong red fluorescence due to the destruction of π-conjugated system. Notably, a good linearity (R^2^ = 0.9961) between the fluorescence intensity at 630 nm and pH in the range of 3.78 to 5.02 was obtained, as well as in the pH range from 5.33 to 7.54 (R^2^ = 0.9321) (Figure 1C,D). Moreover, BP possesses a pKa value of 4.65 related to the oxazolidine switch (Appendix A).

Density functional theory (DFT) calculations were provided to explain the proposed mechanism. As shown in Appendix A, the electrons on the highest occupied orbital (HOMO, −0.1206 eV) and on the lowest unoccupied orbital (LUMO, 0.1584 eV) of BP were all spread in the boron complex part. The electrons on the HOMO of BP**^+^** were also distributed mainly on the boron complex part, while the electrons on the LUMO of BP^+^ located in the axil oxazolidine opening part. Here, we proposed that the quenching of fluorescence in acidic conditions is attributed to the transformation of electrons from the boron complex part to the oxazolidine opening part.

### 3.3. Selectivity Study of BP

To further evaluate the selectivity of BP, we investigated the fluorescence of BP (10 μM) in response to relevant interfering species, including cations, small biomolecules, and reactive oxygen species. As illustrated in Figure 2A, negligible interfering effects on the fluorescence intensity of BP at 630 nm in the presence of all the above species were observed when the pH was 3.78. Additionally, when the pH was 7.54, similar responses were observed, as depicted in Figure 2B. These results demonstrated that the probe possessed excellent selectivity for pH measurement.

### 3.4. Kinetic and Reversible Studies

The time-dependent fluorescence of BP (10 μM) at pH 3.78 and 7.54 was investigated by recording the fluorescence intensity at 630 nm. As described in Figure 2C, after the addition of BP (10 μM) to B-R buffer solutions with 50% MeCN at pH 3.78 and 7.54, the probe rapidly reached equilibrium and remained relatively stable for at least 4 h. Furthermore, the reversibility of this probe was studied in B-R buffer solutions with 50% MeCN. Particularly, this probe functioned well even after 7 cycles when the pH changed from 3.78 to 7.54 (Figure 2D). Therefore, we believe that this probe can serve as a real-time and reversible monitoring of pH.

### 3.5. Cell Imaging with BP at Different pH Values

The aim of this study was to develop an appropriate tool for monitoring pH in biological systems. The MTT study of BP was first assessed in HeLa and U87 cells, which displayed low cytotoxicity at the indicated concentrations, from 5 to 80 μM (Appendix A). To explore the biological application of BP, fluorescent imaging of both HeLa cells and U87 cells was performed at different pH values (3.78, 5.33 and 7.54) in cells treated with 10 μM BP. In Figure 3A, very weak fluorescence was observed in HeLa cells at pH 3.78, whereas significantly enhanced red fluorescence was present in cells treated at pH 5.33 and 7.54. Similar results were also observed in the U87 cells (Figure 3B). The cell imaging results are consistent with the fluorescent properties of BP at different pH values.

### 3.6. Colocalization of Mitochondria and Lysosomes in Living Cells 

In aerobic eukaryotic cells, mitochondria that are energy-producing compartments are central to cell survival and cell death [42,43]. The inner membrane of mitochondria has a proton concentration gradient with a negative charge [44,45]. As a result, small molecules with a positive charge tend to be attracted to the negative potential of the mitochondrial membrane and therefore effectively target mitochondria due to electrostatic attraction [46,47,48]. We subsequently explored the ability of BP to target and lock in mitochondria. As illustrated in Figure 4, the colocalization experiments with MitoTracker Green FM (λ_ex_ = 490 nm and λ_em_ = 513 nm) demonstrated satisfactory targeting ability, with good Pearson’s colocalization coefficients (0.9763 in HeLa cells and 0.8772 in U87 cells). Therefore, BP can localize in mitochondria because it is positively charged in the oxazolidine ring-opening structure.

Lysosomes, which are membrane-surrounded organelles, play a crucial role in the degradation of almost all content within cells [49,50,51]. Lysosomes are digestive compartments in cells that exert their activity at low pH [52]. It has been reported that cyclized indolines can localize in lysosomes in a tertiary amine form that acts as a proton receptor [47,53]. This is consistent with the oxazolidine ring-closed structure, which enlightened the colocalization study of BP toward lysosomes. As illustrated in Figure 5, the colocalization experiments with LysoTracke Green DND-26 (λ_ex_ = 504 nm and λ_em_ = 511 nm) also displayed acceptable tracking ability, with Pearson’s coefficients of 0.8605 in HeLa cells and 0.7697 in U87 cells. Thus, BP has been reported as a tool for successfully localizing lysosomes. Unexpectedly, BP exhibited the properties of simultaneously tracking mitochondria and lysosomes, which were analyzed as molecular tautomerism with specific organelle-targeting structures induced by pH.

To better identify the simultaneous colocalization of mitochondria and lysosomes, BP and two organelle-targeting trackers, MitoTracker Green FM (λ_ex_ = 490 nm and λ_em_ = 513 nm) and LysoTracker Blue (λ_ex_ = 373 nm and λ_em_ = 422 nm), were simultaneously adopted for further colocalization experiments [54]. U87 cells and HeLa cells were pretreated with 10 μM BP, subsequently incubated with MitoTracker Green FM and LysoTracker Blue DND-22. As depicted in Figure 6, BP exhibited strong red fluorescence in U87 and HeLa cells (pH = 7.4). Significantly, the overlay between BP and MitoTracker Green (Figure 6A) displayed that the green fluorescence had a desirable overlap with red fluorescence, whereas the non-merged parts (white circle) coincided well with blue fluorescence from LysoTracker Blue. Coincidentally, a similar phenomenon was also observed in HeLa cells (Figure 6B). Additionally, the corresponding Pearson’s correlation coefficients were 0.9535 and 0.9069 for mitochondria and lysosomes in U87 cells, while the corresponding Pearson’s correlation coefficients were 0.8965 and 0.8487 for mitochondria and lysosomes in HeLa cells. These results more fully proved the simultaneous colocalization of mitochondria and lysosomes by BP.

## 4. Conclusions

In the current study, we developed a BOBPY-oxazolidine-derived pH-responsive probe BP for simultaneously imaging mitochondria and lysosomes. The design of BP was based on pH-dependent oxazolidine rings that were open and close. The oxazolidine ring-open structure could convert to a red emissive ring-closed structure in the presence of a relatively high pH. Remarkably, BP exhibited high stability and reversibility, as well as low cytotoxicity, in HeLa cells and U87 cells. It was successfully applied to visualize the pH alterations in both cell lines. Unconventionally, this probe revealed good colocalization with both mitochondria and lysosomes in living HeLa cells and U87 cells, due to pH-induced molecular tautomerism that forms specific organelle-targeting structures. The studied fluorescent pH probe can simultaneously target mitochondria and lysosomes via pH-induced structure tautomerism, which is an advance of fluorescent probes for real-time cellular visualization. Tracking dual organelles with only one fluorescent probe also provides a new strategy for further developing effective fluorescent tools for exploring organelle interactions. 

## Data Availability

Not applicable.

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
