# Peer review of "Rational Design of Oxazolidine-Based Red Fluorescent pH Probe for Simultaneous Imaging Two Subcellular Organelles"

_biosensors, 2022, doi:10.3390/bios12090696_

Round 1
Reviewer 1 Report
This manuscript describes a near infrared fluorescent probe for pH response. The structure combines with a bodipy moiety and a switchable hemicyanine moiety, these two moieties were linked through a boron complex bonding. The structure is interesting. However, the author didn’t give clear explanation and enough data to demonstrate the mechanism. Author needs to provide more data and major revisions before publishing in this journal
1.Author didn’t clearly claim the mechanism of the response. Although acidic condition can cause ring open of the hemicyanine dye moiety, how does this change enhance the brightness of the BODIPY moiety? Is ring-opening form of hemi-cyanine dye moiety a quencher? To better understand the mechanism, author needs to show us the absorption and emission properties of isolate BP moiety and hemi-cyanine dye moiety (close and open form).
2. In figure 1A, we can see the peak centered at 400 nm decrease with the increase of pH, there should be a new peak come out at same time. Where is this close-form peak?
3. what is pKa of this pH fluorescent probe?
4. Please test the UV light response of the probe.
5. Since hemicyanine dye moiety was reported to be sensitive to solvent, please test the optical properties of BP in different solvent.
6 Is Boron-R complex stable in base pH condition? Will these two moieties detach in base condition? Please add pH response for this probe at pH 8,9 and 10.
Author Response
Referee: 1
Comments: This manuscript describes a near infrared fluorescent probe for pH response. The structure combines with a bodipy moiety and a switchable hemicyanine moiety, these two moieties were linked through a boron complex bonding. The structure is interesting. However, the author didn’t give clear explanation and enough data to demonstrate the mechanism. Author needs to provide more data and major revisions before publishing in this journal
Q1. Author didn’t clearly claim the mechanism of the response. Although acidic condition can cause ring open of the hemicyanine dye moiety, how does this change enhance the brightness of the BODIPY moiety? Is ring-opening form of hemi-cyanine dye moiety a quencher? To better understand the mechanism, author needs to show us the absorption and emission properties of isolate BP moiety and hemi-cyanine dye moiety (close and open form).
Response: We thank the review for this comment.
Density functional theory (DFT) calculations were provided to explain the proposed mechanism. As shown in figure below, the electrons on the highest occupied orbital (HOMO, -0.1206 eV) and on the lowest unoccupied orbital (LUMO, 0.1584 eV) of BP were spread in the boron complex part. The electrons on the HOMO of BP+ were also distributed mainly on the boron complex part, while the electrons on the LUMO of BP+ located in the axil oxazolidine opening part. Here, we proposed that the quenching of fluorescence in acidic condition was attributing to the transformation of electrons from the boron complex part to oxazolidine opening part.
Figure Frontier molecular orbital energy diagram of BP and its opening structure BP+ in vacuum. Transition energies were calculated using the TD-B3LYP method with 6-31G basis sets.
Table Calculated excitation energy (eV), oscillator strengths and major contribution for BP and BP+.
|
Compounds |
Energy Gap (eV) |
f |
Composition |
HOMO (eV) |
LUMO (eV) |
|
BP |
2.3207 |
0.3385 |
HUMO→LUMO 0.6733 |
-0.1206 |
0.1584 |
|
BP+ |
1.9704 |
0.1072 |
HUMO→LUMO 0.7063 |
0.1526 |
0.6844 |
Additionally, the absorption and emission spectra of isolate BP moiety and hemi-cyanine dye moiety (close and open form) were shown in figure below.
Figure Absorption (A) and emission (B) spectra of hemi-cyanine moiety (close and open form) and BOBPY-CHO.
Q2. In figure 1A, we can see the peak centered at 400 nm decrease with the increase of pH, there should be a new peak come out at same time. Where is this close-form peak?
Response: We thank the review for this comment.
N2O-type benzopyrromethene boron complexes (BOBPYs) were first developed by Jiao and coworkers [1], in which different boronic acid derivatives were substituted in the axial position. The results demonstrated that different substitutions in the axial position influenced the absorption wavelength very little. So, the absorption peak of BP at 620 nm did not change significantly in both ring opening structure and ring closing structure (Figure 1A), while the absorption peak centered at 400 nm changed significantly due to oxazolidine ring opening and closing.
Ref. [1] Chen, N.; Zhang, W.; Chen, S.; Wu, Q.; Yu, C.; Wei, Y.; Xu, Y.; Hao, E.; Jiao, L., Sterically Protected N2O-Type Benzopyrromethene Boron Complexes from Boronic Acids with Intense Red/Near-Infrared Fluorescence. Org. Lett. 2017, 19, (8), 2026-2029.
Q3. what is pKa of this pH fluorescent probe?
Response: We thank the review for this valuable comment. The pKa of BP in this study was calculated as 4.65 according to the following methods.
The fluorometric titration as a function of pH was obtained fluorescence spectra. The equation below was used to calculate the pKa value of probe BP [2].
F=(Fmin[H+]n + FmaxKa)/(Ka+[H+]n)
The expression of the steady-state fluorescence intensity F as a function of the proton concentration has been extended for the case of n: complex between H+ and a fluorescent dye. Where Fmin and Fmax are the fluorescence intensities at maximal and minimal H+ concentrations, respectively. n is apparent stoichiometry of H+ binding to the probe BP. Nonlinear fitting of equation expressed above to the fluorescence titration data was plotted as a function of H+ concentration.
Figure Plot curve of fluorescence intensity at 620 nm of 10 μM probe BP versus pH. The information of the curve: Model, Boltzmann. Equation, y=A2+(A1-A2)/(1+exp((x-x0)/dx)). pKa=x0.
Ref. [2] Mazi, W.; Yan, Y.; Zhang, Y.; Xia, S.; Wan, S.; Tajiri, M.; Luck, R. L.; Liu, H., A near-infrared fluorescent probe based on a hemicyanine dye with an oxazolidine switch for mitochondrial pH detection. J. Mater. Chem. B 2021, 9, (3), 857-863.
Q4. Please test the UV light response of the probe.
Response: We thank the review for this comment. The UV light response of BP in the mixture of MeCN and B-R buffer solution (v/v = 1:1, pH = 7.54) were recorded at indicated times. There present no significant changes in absorption (figure below), which indicated that BP is stable under the illumination of UV light.
Figure The absorption spectra of BP (10 μM) in the mixture of MeCN and B-R buffer solution (v/v = 1:1, pH = 7.54) illuminated by UV light at indicated times.
Q5. Since hemicyanine dye moiety was reported to be sensitive to solvent, please test the optical properties of BP in different solvent.
Response: We thank the review for this comment. There are no significant changes of absorption and emission spectra of BP in different solvents. However, BP has bad solubility in H2O resulting in aggregation which can be observed from following figure A. Therefore, BP displayed almost no fluorescence in H2O (following figure B).
Figure Absorption and emission spectra of BP in different solvents.
Q6. Is Boron-R complex stable in base pH condition? Will these two moieties detach in base condition? Please add pH response for this probe at pH 8,9 and 10.
Response: We thank the review for this comment. As shown in following figure A, BOBPY-CHO is stable at different pH (from 3.78 to 10.38). Besides, BP is also stable at pH 8.36, 9.15 and 10.38, which indicated that two moieties did not detach in base condition (following figure B-C). The pH response for BP at pH 8.36, 9.15 and 10.38 is similar to that at 7.54.
Figure (A) Fluorescence responses of BOBPY-CHO (10 μM) in B-R buffer with 50% MeCN at different pH values (from 3.78 to 10.38). Absorption spectra (B) and fluorescent spectra (C) of BP (10 μM) in the mixture of MeCN and B-R buffer solution (v/v = 1:1) at pH 7.54, 8.36, 9.15, 10.38.

Reviewer 2 Report
In this manuscript, Zhang and co-workers are reporting an interesting fluorescent probe for simultaneous visualization of mitochondria and lysosomes for live cell imaging applications. Although the development of dual sensitive probe are not entirely novel, the developed probe seems to be interesting and therefore I believe it is a suitable topic to be published in Biosensors. However the presented format of the manuscript is raising several doubts. Therefore, I recommend authors to carefully revise their manuscript by following my comment. Also, i would like to emphasize that "dual selectivity" of fluorescent probes is a very sensitive topic and therefore all experimental data must be statistically evaluated properly.
(1). Manuscript contains multiple grammatical and textual errors, therefore authors should revise it accordingly.
(2). Authors should clearly highlight the importance of the dual selectivity of a probe. Some previously reported probes with this property, must be cited: J. Mater. Chem. B, 2018,6, 1716-1733; J Fluoresc 31, 1227–1234 (2021); J. Am. Chem. Soc. 2015, 137, 18, 5930–5938; J. Am. Chem. Soc. 2016, 138, 38, 12368–12374.
(3). What is the rationale of the design of this fluorophore to expect dual selectivity? or was this a coincidence?
(4). Authors should clearly tabulate all spectroscopic properties of the probe in different solvents which must include, absorbance, emission, Stokes' shift, molar absorptivity and the quantum yield.
(5). For HRMS data, authors must provide ppm error.
(6). Why authors use such a high staining concentration as 10 uM. It seems to be authors are overloading the cells with probe. In situations like this, it is obvious to observe multiple organelle selectivity. Therefore, in order to rule out this is a concentration effect, colocalization data must be performed at least with two lower concentrations (i.e. 2 uM and 5 uM), in order to rule out this.
(7). Microscopy images presented in the submitted manuscript is in poor quality. Authors must submit high resolution images alongside with digital zoomed-in images in order to clearly see probe's imaging pattern.
(8). In all microscopy imaging figure captions, authors should clearly mention, what is the staining concentration for each dye, which lasers used for the excitation and how does the emission was collected. These information is vital.
(9). Authors should clearly mention how co-localization experiments were conducted, how the analyze the overlap coefficients in details. especially it is important to mention how many replicates and how many cells were used. These data must be reported with error bars graphically.
(10). Without seeing colocalization images and calculated overlap coefficients for the probe with lower concentration, the homogeneity of this work is questionable.
Author Response
Referee: 2
Comments: In this manuscript, Zhang and co-workers are reporting an interesting fluorescent probe for simultaneous visualization of mitochondria and lysosomes for live cell imaging applications. Although the development of dual sensitive probe is not entirely novel, the developed probe seems to be interesting and therefore I believe it is a suitable topic to be published in Biosensors. However, the presented format of the manuscript is raising several doubts. Therefore, I recommend authors to carefully revise their manuscript by following my comment. Also, i would like to emphasize that "dual selectivity" of fluorescent probes is a very sensitive topic and therefore all experimental data must be statistically evaluated properly.
Q1. Manuscript contains multiple grammatical and textual errors; therefore, authors should revise it accordingly.
Response: We thank the review for this comment. We have carefully corrected the grammatical and textual errors and the revised parts were marked in the manuscript.
Q2. Authors should clearly highlight the importance of the dual selectivity of a probe. Some previously reported probes with this property, must be cited: J. Mater. Chem. B, 2018,6, 1716-1733; J Fluoresc 31, 1227–1234 (2021); J. Am. Chem. Soc. 2015, 137, 18, 5930–5938; J. Am. Chem. Soc. 2016, 138, 38, 12368–12374.
Response: We thank the review for this comment.
For acidic lysosomes and alkalescent mitochondria, real-time monitoring by only one fluorescent pH probe can obtain deep insight into the more complicated and important cellular or subcellular activities, as well as their interactions. Many recent studies have made great success in exploring the contact between lysosomes and mitochondria, as well as useful biomolecule HClO containing [31-34].
Added references in the manuscript:
- Abeywickrama, C. S.; Baumann, H. J.; Pang, Y., Simultaneous Visualization of Mitochondria and Lysosome by a Single Cyanine Dye: The Impact of the Donor Group (-NR2) Towards Organelle Selectivity. J. Fluoresc. 2021, 31, (5), 1227-1234.
- Yuan, L.; Wang, L.; Agrawalla, B. K.; Park, S.-J.; Zhu, H.; Sivaraman, B.; Peng, J.; Xu, Q.-H.; Chang, Y.-T., Development of Targetable Two-Photon Fluorescent Probes to Image Hypochlorous Acid in Mitochondria and Lysosome in Live Cell and Inflamed Mouse Model. J. Am. Chem. Soc. 2015, 137, (18), 5930-5938.
- Liu, Y.; Zhou, J.; Wang, L.; Hu, X.; Liu, X.; Liu, M.; Cao, Z.; Shangguan, D.; Tan, W., A Cyanine Dye to Probe Mitophagy: Simultaneous Detection of Mitochondria and Autolysosomes in Live Cells. J. Am. Chem. Soc. 2016, 138, (38), 12368-12374.
- Abeywickrama, C. S.; Wijesinghe, K. J.; Stahelin, R. V.; Pang, Y., Bright red-emitting pyrene derivatives with a large Stokes shift for nucleus staining. Chem. Commun. 2017, 53, (43), 5886-5889.
Q3. What is the rationale of the design of this fluorophore to expect dual selectivity? or was this a coincidence?
Response: We thank the review for this comment. We designed BP for simultaneous imaging lysosomes and mitochondria.
As well known, the inner membrane of mitochondria has a proton concentration gradient with a negative charge [44,45]. As a result, small molecules with a positive charge tend to be attracted to the negative potential of the mitochondrial membrane and therefore effectively targeting mitochondria due to electrostatic attraction [46-48]. Here, oxazolidine ring opening structure of BP is the positive charge structure for potential mitochondria targeting. Additionally, it has been reported that cyclized indolines can localize in lysosomes in a tertiary amine form that acts as a proton receptor [47,53]. This is consistent with the oxazolidine ring-closed structure of BP. Collectively, we designed BP for simultaneous imaging lysosomes and mitochondria.
Reference in the manuscript:
- Kroemer, G.; Galluzzi, L.; Brenner, C., Mitochondrial Membrane Permeabilization in Cell Death. Physiol. Rev. 2007, 87, (1), 99-163.
- Hoye, A. T.; Davoren, J. E.; Wipf, P.; Fink, M. P.; Kagan, V. E., Targeting Mitochondria. Acc. Chem. Res. 2008, 41, (1), 87-97.
- Zorova, L. D.; Popkov, V. A.; Plotnikov, E. Y.; Silachev, D. N.; Pevzner, I. B.; Jankauskas, S. S.; Babenko, V. A.; Zorov, S. D.; Balakireva, A. V.; Juhaszova, M.; Sollott, S. J.; Zorov, D. B., Mitochondrial membrane potential. Anal. Biochem. 2018, 552, 50-59.
- Park, S. J.; Juvekar, V.; Jo, J. H.; Kim, H. M., Combining hydrophilic and hydrophobic environment sensitive dyes to detect a wide range of cellular polarity. Chem. Sci. 2020, 11, (2), 596-601.
- Zhu, H.; Fan, J.; Du, J.; Peng, X., Fluorescent Probes for Sensing and Imaging within Specific Cellular Organelles. Acc. Chem. Res. 2016, 49, (10), 2115-2126.
- Chen, W.; Gao, C.; Liu, X.; Liu, F.; Wang, F.; Tang, L. J.; Jiang, J. H., Engineering Organelle-Specific Molecular Viscosimeters Using Aggregation-Induced Emission Luminogens for Live Cell Imaging. Anal. Chem. 2018, 90, (15), 8736-8741.
Q4. Authors should clearly tabulate all spectroscopic properties of the probe in different solvents which must include, absorbance, emission, Stokes' shift, molar absorptivity and the quantum yield.
Response: We thank the review for this comment. The spectroscopic properties of BP in different solvents were shown in the following table.
|
Solvents |
λabs (nm) |
λem (nm) |
Stokes shift (nm) |
Absorptivity a |
Quantum yield b |
|
MeCN |
621 |
630 |
9 |
51600 |
0.58 |
|
MeOH |
622 |
633 |
11 |
55800 |
0.56 |
|
THF |
625 |
629 |
4 |
58800 |
0.55 |
|
Tol |
630 |
636 |
6 |
56000 |
0.53 |
Note: a, Molar extinction coefficients are in the maximum of the highest peak. b, Quantum yields were detected by fluorescence spectrophotometer (FluoroMax-4, Horiba, Japan).
Q5. For HRMS data, authors must provide ppm error.
Response: We thank the review for this comment. The HRMS data was revised in the manuscript and supplementary materials.
Q6. Why authors use such a high staining concentration as 10 uM. It seems to be authors are overloading the cells with probe. In situations like this, it is obvious to observe multiple organelle selectivity. Therefore, in order to rule out this is a concentration effect, colocalization data must be performed at least with two lower concentrations (i.e. 2 uM and 5 uM), in order to rule out this.
Response: We thank the review for this valuable comment. 2 μM and 5 μM of BP were used to perform the colocalization experiments in U87 cells and HeLa cells. As shown in following figure, 2 μM and 5 μM of BP displayed satisfactory targeting ability to lysosomes and mitochondria in these two cell lines. Pearson’s colocalization coefficients were similar to that of cells treated with 10 μM BP, which indicated that the dual targeting ability is not attributed to overload the cells with BP.
Figure Colocalization experiments of BP with MitoTracker Green FM and LysoTracker Green DND-26 in U87 cells (A) and HeLa cells (B). Lyso/Mito channel: λex = 488 nm, λem = 490-530 nm; BP channel: λex = 561 nm, λem = 600-750 nm.
Q7. Microscopy images presented in the submitted manuscript is in poor quality. Authors must submit high resolution images alongside with digital zoomed-in images in order to clearly see probe's imaging pattern.
Response: We thank the review for this valuable comment. High resolution images were provided in Figure 6 to clearly observe the imaging pattern between BP and commercial dyes.
Figure 4 and Figure 5 displayed the colocalization of BP in U87 cells and HeLa cells with MitoTracker Green FM (λex = 490 nm and λem = 513 nm) and LysoTracker Green DND-26 (λex = 504 nm and λem = 511 nm), respectively. To better identified the simultaneous colocalization of mitochondria and lysosomes, BP, MitoTracker Green FM (λex = 490 nm and λem = 513 nm) and LysoTracker Blue (λex = 373 nm and λem = 422 nm), were simultaneously adopted for further colocalization experiments (Figure 6).
U87 cells and HeLa cells were pre-treated with 10 μM BP, subsequently incubated with MitoTracker Green FM and LysoTracker Blue DND-22. As depicted in Figure 6, BP exhibited strong red fluorescence in U87 cells and HeLa cells (pH = 7.4). Significantly, the overlay between BP and Mito-Tracker Green (Figure 6A) displayed that the green fluorescence had desirable overlap with red fluorescence, whereas the non-merged parts (white circle) was well coincided with blue fluorescence from LysoTracker Blue. Coincidentally, the similar phenomenon could also be observed in HeLa cell (Figure 6B). Additionally, the corresponding Pear-son’s correlation coefficients were 0.9535 and 0.9069 for mitochondria and lysosomes in U87 cells, while the corresponding Pearson’s correlation coefficients were 0.8965 and 0.8487 for mitochondria and lysosomes in HeLa cells.
Figure 6. Colocalization experiments of BP with MitoTracker Green FM and LysoTracker Blue DND-22 in U87 cells (A) and HeLa cells (B). Blue channel: λex = 405 nm, λem = 405-450; Green channel: λex = 488 nm, λem = 490-530 nm; Red channel: λex = 561 nm, λem = 600-750 nm.
Q8. In all microscopy imaging figure captions, authors should clearly mention, what is the staining concentration for each dye, which lasers used for the excitation and how does the emission was collected. These information is vital.
Response: We thank the review for this comment. The detailed information about the commercial dyes and BP was described in the Section “2.5 Cell Treatment and Imaging” from the manuscript. Besides, the excitation and emission were also added in the manuscript.
The two cell types were respectively pretreated with 20 μM Mito-Tracker Green FM (λex = 490 nm and λem = 513 nm) and 20 μM LysoTracker Blue DND-22 (λex = 373 nm and λem = 422 nm) for 30 min. After washing with PBS buffer (pH=7.4) three times, the cells were subsequently incubated with 10 μM BP for 1h. Before imaging, the culture medium was discarded, and the cells were washed with PBS buffer (pH=7.4) three times and then fixed.
For cell imaging, the channels are blue channel (λex=405 nm, λem=405-450), green channel (λex=488 nm, λem=490-530 nm) and red channel (λex=561 nm, λem=600-750 nm).
Q9. Authors should clearly mention how co-localization experiments were conducted, how the analyze the overlap coefficients in details. especially it is important to mention how many replicates and how many cells were used. These data must be reported with error bars graphically.
Response: We thank the review for this comment. The details about the colocalization experiments were described in Section “2.5 Cell Treatment and Imaging” from the manuscript.
After 90% confluence, HeLa cells and U87 cells were seeded in culture dishes at a density of 1×105 for 24 h. For colocalization of lysosomes and mitochondria, the cells were pretreated with 20 μM MitoTracker Green FM (λex = 490 nm and λem = 513 nm) or 20 μM LysoTracker Green DND-26 (λex = 504 nm and λem = 511 nm) for 30 min and then washed with PBS buffer (pH = 7.4) three times. After that, the cells were treated with 10 μM BP for 1 h. Before imaging, the culture medium was discarded, and the cells were washed with PBS buffer (pH = 7.4) three times and then fixed. Images were also collected by confocal fluorescence microscopy (Carl Zeiss LSM710, German). Data from three independent experiments were then analyzed using Image J.
In addition, the two cell types were respectively pretreated with 20 μM Mito-Tracker Green FM (λex = 490 nm and λem = 513 nm) and 20 μM LysoTracker Blue DND-22 (λex = 373 nm and λem = 422 nm) for 30 min. After washing with PBS buffer (pH = 7.4) three times, the cells were subsequently incubated with 10 μM BP for 1h. Before imaging, the culture medium was discarded, and the cells were washed with PBS buffer (pH = 7.4) three times and then fixed. Images were also collected by confocal fluorescence microscopy (Carl Zeiss LSM710, German). Data from three independent experiments were then analyzed using Image J.
The following figure exhibited the data of Pearson’s colocalization coefficients from three independent experiments with error bars.
Q10. Without seeing colocalization images and calculated overlap coefficients for the probe with lower concentration, the homogeneity of this work is questionable.
Response: We thank the review for this valuable comment. 2 μM and 5 μM of BP were used to perform the colocalization experiments in U87 cells and HeLa cells. As shown in following figure A, 2 μM and 5 μM of BP displayed satisfactory targeting ability to lysosomes and mitochondria in these two cell lines. Pearson’s colocalization coefficients were similar to that of cells treated with 10 μM BP.
Figure Colocalization experiments of BP with MitoTracker Green FM and LysoTracker Green DND-26 in U87 cells (A) and HeLa cells (B). Lyso/Mito channel: λex = 488 nm, λem = 490-530 nm; BP channel: λex = 561 nm, λem = 600-750 nm.

Round 2
Reviewer 1 Report
The responses are all good to answer reporter 1's questions. I think it can be accepted for publication in Biosensers.
Reviewer 2 Report
Authors have reasonably answered majority of my comments. Therefore, I recommend accepting in the current form.